# Water filtration using softwood membranes provides a nature-based solution for nanoplastic removal
Alice Pradel [1,4] ✉, Maximilian Ritter[2,3,4], Wenqing Yan[2] & Denise M. Mitrano[1]

Nanoplastic (NP) contamination in water intended for human consumption will require efficient, environmentally friendly, and cost-effective NP-removal methods. This study evaluates the use of native wood membranes to filter NPs from freshwater using a pressure-driven process. The performance of the filter was assessed for different wood species, membrane thicknesses and filtration cycles, by measuring NP retention efficiency and NP deposition in the wood membranes. Retention efficiency varied widely between species. After a single filtration cycle, spruce, a common softwood species, reduced NP concentrations by approximately 90%, while poplar, a hardwood, only reduced NP concentrations by ~20%. These differences are linked to microstructural differences between soft- and hardwood tissues. Increased membrane thickness and repeated filtration further enhanced NP retention in all conditions. This work demonstrates that native softwood membranes (i.e., without chemical modification) can be used in low-pressure filtration systems to remove NPs.

Despite access to safe drinking water being fundamental to human well-being, contaminants such as pharmaceuticals, pesticides, perfluorinated compounds, and plastic particles are frequently found in drinking water[1–3]. These contaminants are generally found at concentrations below acute toxicity thresholds, but the effects of chronic exposure are unknown. Nanoplastics (NPs, 1 nm–1 µm) constitute one of the contaminants that should ideally be removed from drinking water prior to human consumption to ensure its safety. NPs are widely distributed in the environment and can be found in surface[4] and groundwaters[5] that are then used as drinking water sources. In addition, a recent study suggests that NPs can be released from distribution pipelines after treatment[6]. The occurrence of NPs in raw and treated drinking water is less documented than the occurrence of microplastics (MPs, 1 µm to 5 mm)[2] due to the analytical challenges involved in quantifying NPs compared to MPs[7,8]. However, the frequent contamination by MPs in drinking water suggests that NPs may be abundant as well[2]. In fact, a first quantification has measured 3.9 ± 0.88 µg/L of polystyrene NPs in tap water[9]. Therefore, to reduce exposure to contaminants, water treatment technologies capable of removing colloidal contaminants, including NPs, may need to be applied either at drinking water treatment plants or at household levels, depending on the source of contamination.

Drinking water treatment plants harvest freshwater from surface or underground sources and remove microbial and chemical contaminants by combining physical and chemical processes[10]. Physical processes include filtration and coagulation, and are most commonly used for the removal of suspended solids and thus may also be an appropriate approach for NP removal. Activated carbon filters were found to be effective at removing NPs at the laboratory-scale[11,12]. However, when scaling up to drinking water treatment plants, activated carbon filtration, like rapid sand filtration, had limited effectiveness in removing NPs (36% and 58% removal, respectively)[13]. The presence of biofilm in slow sand filtration was the most effective condition for removing NPs (>99% removal). However, the presence of biofilm is not constant throughout the lifecycle of slow sand filters, thereby potentially reducing NP removal efficiency. Furthermore, in the case of activated carbon filters, the sustainability of the water treatment process is reduced due to the use of energy-intensive pyrolysis. Current pressure-driven membrane technologies, such as ultra-filtration, nano-filtration, and reverse osmosis, can retain colloidal contaminants, such as NPs, by size exclusion[14–16]. However, as they are composed of synthetic polymers, their production and subsequent use may release NPs, and so the use of alternative, bio-sourced, biodegradable materials could alleviate some of these concerns. For example, a biosourced nanofibre hydrogel has shown promise to remove NPs larger than 30 nm[17]. Other novel filtration materials are being developed, such as graphene oxides[18] and metal-organic composites[19], but these may have limited applicability due to the complexity of their production process, the high cost of the materials, and, as for

[1]Environmental Systems Science Department, ETH Zurich, Zurich, Switzerland. [2]Institute for Building Materials, ETH Zürich, Zürich, Switzerland. [3]Cellulose and Wood Materials, Swiss Federal Laboratories for Materials Science and Technology, Dübendorf, Switzerland. [4]These authors contributed equally: Alice Pradel, Maximilian Ritter. ✉e-mail: pradel@cerege.fr

classical membranes, their non-biodegradable nature. Removal of NPs by coagulation was proven to be effective when combining both cationic coagulants and polymeric flocculants[20]. However, the synthetic flocculants which are generally used (e.g., polyacrylamide), may themselves release toxic residual monomers or oxidation byproducts[21]. Finally, chemical processes of drinking water treatment plants, found in certain water treatment plants, such as ozonation, UV-treatment, and chlorination, can minimize microbial and dissolved contaminants (e.g., pharmaceuticals, microbial toxins, etc.), but are largely ineffective in treating NPs due to their polymeric nature[13]. Overall, these recent studies highlight the need for methods to efficiently remove NPs at low costs and with low environmental impact.

Wood is a potentially efficient filtering material due to its hierarchical, porous structure. Its environmental and economic costs may be low due to its renewable nature and its abundance[22]. The wood structure is optimised for capillary-driven water transport and mechanical strength in living trees. These water transport properties can also be exploited for functionalised wood applications[23–25]. At the lab scale, it has been demonstrated that native wood membranes can remove over 99.9% of suspended micron-sized bacteria[26]. Inspired by this previous success, this study explores the potential of native (i.e., non-modified) wood cross-sections as filters for the removal of NPs suspended in freshwater.

To assess the impact of different wood microstructures, we selected spruce and poplar as representative examples of softwood and hardwood, respectively. On one hand, softwoods primarily consist of tracheids, elongated cells with limited length. In spruce, tracheid length varies depending on age and tree position, but for non-juvenile trees, it generally ranges between 2 and 5 mm, with diameters ranging from 20 to 50 μm[27,28]. Tracheids are connected to each other via bordered pits, facilitating capillary water transport between individual cells. On the other hand, hardwoods rely on vessels for water transport and fibers for structural support. Vessels consist of stacked vessel elements that lose their end walls, forming continuous, hollow tubes with diameters significantly larger than softwood

tracheids. This structure enables more efficient and rapid water transport compared to the purely tracheid-based transport system found in softwoods[29].

Laboratory experiments were performed to assess the NP filtration capacity of cross sections of native Norway spruce (*Picea abies*), a common softwood, and Poplar (*Populus nigra*), a common hardwood. Due to the challenges associated with detecting NPs in organic matrices, we used model NPs doped with palladium (Pd-NP)[30]. The metal functioned as a conservative tracer throughout the experimental system and was quantified by inductively coupled plasma mass spectrometry (ICP-MS). The Pd-NPs were dispersed in artificial freshwater (NP feed suspension) and subsequently filtered through wood membranes of 10 mm or 20 mm thickness, for one or three filtration cycles. Filtration capacity was assessed by measuring the reduction in Pd-NP concentration eluted, as well as the mass of Pd-NP deposited inside the wood membranes. This allowed us to assess the effect of tree species, wood membrane thickness, and number of filtration cycles on Pd-NP retention in wood membranes. As such, this work advances the development of efficient methods to filter NPs and potentially other colloidal contaminants, with reduced economic cost and environmental impact.

## Results and discussion

For membranes to be effective NP filters, they must retain NPs to produce uncontaminated water. We anticipated that the anatomical differences in the structures of poplar (hardwood) and spruce (softwood) would influence their capacity to retain Pd-NPs (Fig. 1a). To test this, we used a filtration setup composed of a wood membrane inserted into a syringe (Fig. 1b). The negatively-charged Pd-NPs were dispersed in artificial freshwater. In addition to comparing the efficiency differences between wood species, we assessed how operational conditions of the system, including increasing the thickness of the wood membrane and recirculating the Pd-NP suspension through the same membrane, would alter Pd-NP retention over time.

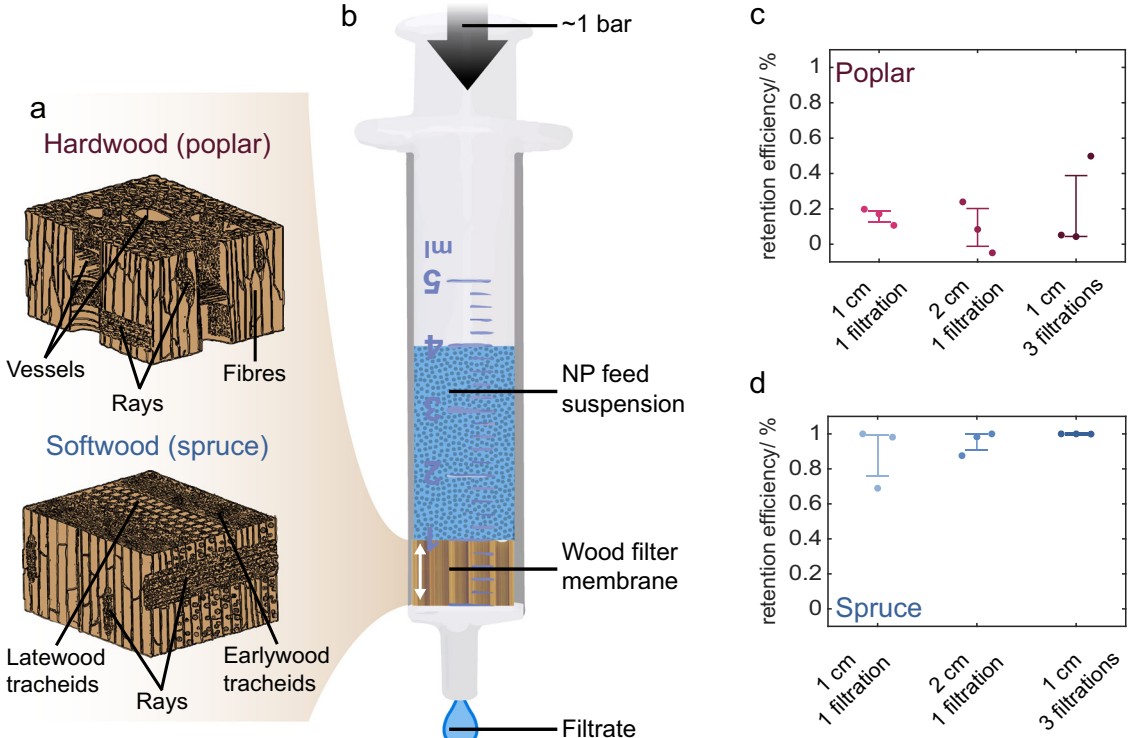

**Fig. 1 | Native wood membrane filters and their capacity to retain Pd-NPs.** **a** Schematics of the hardwood and softwood structure, highlighting the different cell types. **b** Overview of the pressure-driven wood membrane filtration setup, featuring a syringe and a 10 or 20 mm thick wood filter membrane. The arrow indicates the wood fibre direction inside the membrane. Retention efficiency of Pd-NP in **c** poplar and **d** spruce membranes of various thicknesses and filtration cycles (*n* = 3, error bar: lower indicated 25th and upper 75th percentile).

## Wood species is the strongest predictor of Pd-NP retention efficiency

The retention efficiency of Pd-NPs, obtained by dividing the change in Pd-NP concentration in the effluent ($C_0$-C) by the initial concentration ($C_0$), provided a measure of the capacity of Pd-NP to be retained by wood membranes (Fig. 1c, d). The wood species significantly affected the retention efficiency of Pd-NPs in the filters. Poplar membranes showed retention efficiencies of approximately 20%, irrespective of membrane thickness and number of filtration cycles (Fig. 1c). In contrast, spruce membranes had a higher capacity to retain Pd-NP, as the 10 mm thick membranes already exhibited average retention efficiencies rates of 89% with 1 filtration cycle and almost 100% after 3 filtration cycles (Fig. 1d). As expected for a natural material, the retention efficiency fluctuated between individual specimens. Despite this variability, significant differences were observed for hardwood and softwood membranes. We hypothesize that the low retention efficiency of poplar was due to the fact that the vessels present in the hardwood microstructure allowed an unimpeded water flow through the wood membrane due to the absence of end walls. Therefore, the Pd-NP were suspected to flow through the vessels without being intercepted by wood surfaces or constrictions along the flow path. In contrast, in spruce, due the tracheids' finite length, the water must flow transversely to the dominant flow path, through the pits that connect individual tracheids. The openings of these pits, which are 3–7 μm in diameter, are smaller than the tracheid diameter (20–50 μm). As a result, they act as constriction zones, increasing the likelihood of interception and deposition on the wood surface[31] (Fig. S2). In this case, Pd-NPs are not retained by size exclusion (i.e., pores smaller than the colloid size), but rather by interception around the pit. This occurs due to the change in streamline direction at the constriction formed by the pit. Such interception at constrictions (i.e., pore throats) with diameters larger than the colloid is suspected to occur in packed-bed filters[32,33] though it cannot be observed directly. However, this process has been observed in microfluidic filters[34–36].

These findings are further supported by observations that with the same applied pressure, spruce and poplar membranes showed different water fluxes. Based on the filtration volume (3 mL), membrane area (154 mm²), and approximate filtration durations (57 s to 5 min), the water flux values were 608 L m$^{-2}$ h$^{-1}$ bar$^{-1}$ for poplar after 1 or 3 filtrations, and 195 and 117 L m$^{-2}$ h$^{-1}$ bar$^{-1}$ for spruce after 1 and 3 filtrations, respectively. Corresponding flow rates were 1.2 m h$^{-1}$ for poplar, and 0.4 m h$^{-1}$ and 0.2 m h$^{-1}$ for spruce. For reference, the flow rate for poplar is slightly above that of typical slow sand filtration but considerably lower than rapid sand filtration (7–20 m h$^{-1}$) in a conventional drinking water treatment plant, while the flow rate for spruce is in the range of slow sand filtration (<0.3 m h$^{-1}$)[37]. These values, obtained for spruce tracheids of 20–50 μm diameter and poplar vessels of about 150 μm diameter, will vary between tree species and tree growth conditions. Previous studies have also demonstrated the influence of wood anatomy on retention of suspended matter. Wood pits have been shown to play a pivotal role in retaining bacteria[26]. Furthermore, a study on nanoparticle impregnation showed that $SiO_2$ nanoparticles were less easily retained through vessels of beech wood (hardwood) than tracheids of pine wood (softwood)[38].

## Deposition profiles of Pd-NPs in the wood membranes reveal the role of wood anatomy

To understand where Pd-NPs were retained in the wood membranes, the top 5 mm and the remaining bottom part of the membrane were separated and individually analysed by ICP-MS to obtain deposition profiles (Fig. 2a, b). Poplar's low retention efficiency translated to a low Pd-NP deposition inside the membrane (Fig. 2a). Indeed, the top 5 mm of poplar membranes contained, on average, only 8 and 6% of the total mass of Pd-NPs injected after one and three filtration cycles. A slight trend towards increased retention in the first 5 mm of the membrane compared to the bottom section was observed, as frequently seen in other porous media. This is due to the fact that the amount of dispersed Pd-NPs that remained to be deposited decreased as the membrane depth increased[33,34,39]. As expected,

the Pd-NP concentration in spruce membranes was significantly higher, especially in the top 5 mm, which contained on average 68% of the total Pd-NPs present in the feed suspension after a single filtration cycle and 78% Pd-NPs after three filtration cycles (Fig. 2b). The bottom part of the spruce membranes contained a significantly lower concentration of Pd-NPs compared to the top. This explains why increasing the thickness from 10 to 20 mm did not yield a significant increase in the retention efficiency (Fig. 1d). Due to technical issues, we could not obtain the data for Pd-NPs deposition in the 20 mm spruce samples.

These deposition profiles obtained by measuring Pd-NPs in cross sections of the wood membrane are in agreement with scanning electron microscope (SEM) micrographs taken at different depths of a spruce membrane (Fig. 2c–k). These SEM micrographs show the tangential section of a 10 mm spruce membrane that was used for 1 filtration cycle. As a reference, the Pd-NPs and the unused spruce membrane are shown in Figs. S1 and S2. SEM micrographs taken at the very top of a used spruce membrane (representative micrographs shown in Fig. 2c–e) reveal numerous areas with significant Pd-NPs deposition close to the pits, but also on the cell wall where there are no pits (Fig. S3). Figure 2f–h shows representative SEM micrographs just below the very top (still within the top 5 mm) and highlights the widespread deposition of Pd-NPs on the cell wall. Micrographs of wood pits on the same height (Fig. S4) show a strong accumulation of Pd-NPs around the pits. This indicates that the pits play a pivotal role in the filtering mechanism of the wood membranes, in line with the findings of previous reports on wood-based filters[26,38]. SEM micrographs from the bottom of the spruce membrane (representative examples shown in Fig. 2i–k) reveal no deposited NPs. SEM analysis, therefore, supports our hypothesis that deposition around the wood pits causes the retention of Pd-NPs in softwood membranes. We attempted to confirm the presence of deposited Pd-NPs in the wood membranes by measuring Pd by energy dispersive X-ray spectroscopy (EDS), as done previously[30], but this was not achievable since the high acceleration voltage necessary for obtaining a Pd signal resulted in beam damage of the wood.

## Native wood membranes as a promising material to filter colloidal contaminants

To contextualize the performance of spruce wood membranes, we compare Pd-NPs retention in native spruce i) to retention of NPs by various filtration systems (Fig. 2l), and ii) to retention of various contaminants by wood membranes (Fig. 2m). On one hand, native spruce membranes outperformed many ultrafiltration, microfiltration, and packed bed filtration systems[15,32,40]. Spruce membranes achieved equal retention efficiency as ultrafiltration membranes, but at a significantly higher flow rate[40]. They could achieve higher retention efficiency than filter microfiltration membranes (0.4 μm)[40] and sand filters[32], and they could also achieve higher flow rates. Finally, nanofibrous membranes composed of synthetic polymers also achieve retention efficiencies up to approximately 0.9, but these had a lower flux compared to spruce membranes, and require energy- and material-intensive preparations[15] (Fig. 2l). On the other hand, the filtration of NPs measured in our work falls within the ranges of retention efficiencies and fluxes observed for other woods (native or functionalized) and other contaminants (dissolved or bacterial; Fig. 2m). To achieve over 90 % retention efficiencies of organic molecules (methylene blue) and metallic ions (copper), wood membranes needed to be functionalized[41,42]. Filtration of methylene blue by carbonized wood achieved the highest combination of retention efficiency and flux[41]. Bacteria were the only particulate matter whose filtration was studied[26,43]. Results showed that both functionalized and native wood achieved well above 90% retention of bacteria[26,43], but the fluxes were significantly lower than for methylene blue in balsa wood[41]. At the same filtration pressure, the flux in both our spruce and poplar membranes was lower than the flux in bacterial filtration studies. Furthermore, spruce membranes, which had a higher retention efficiency of Pd- NP compared to poplar membranes, also had a lower flux compared to poplar membranes (Fig. 2m). This suggests that a low wood membrane

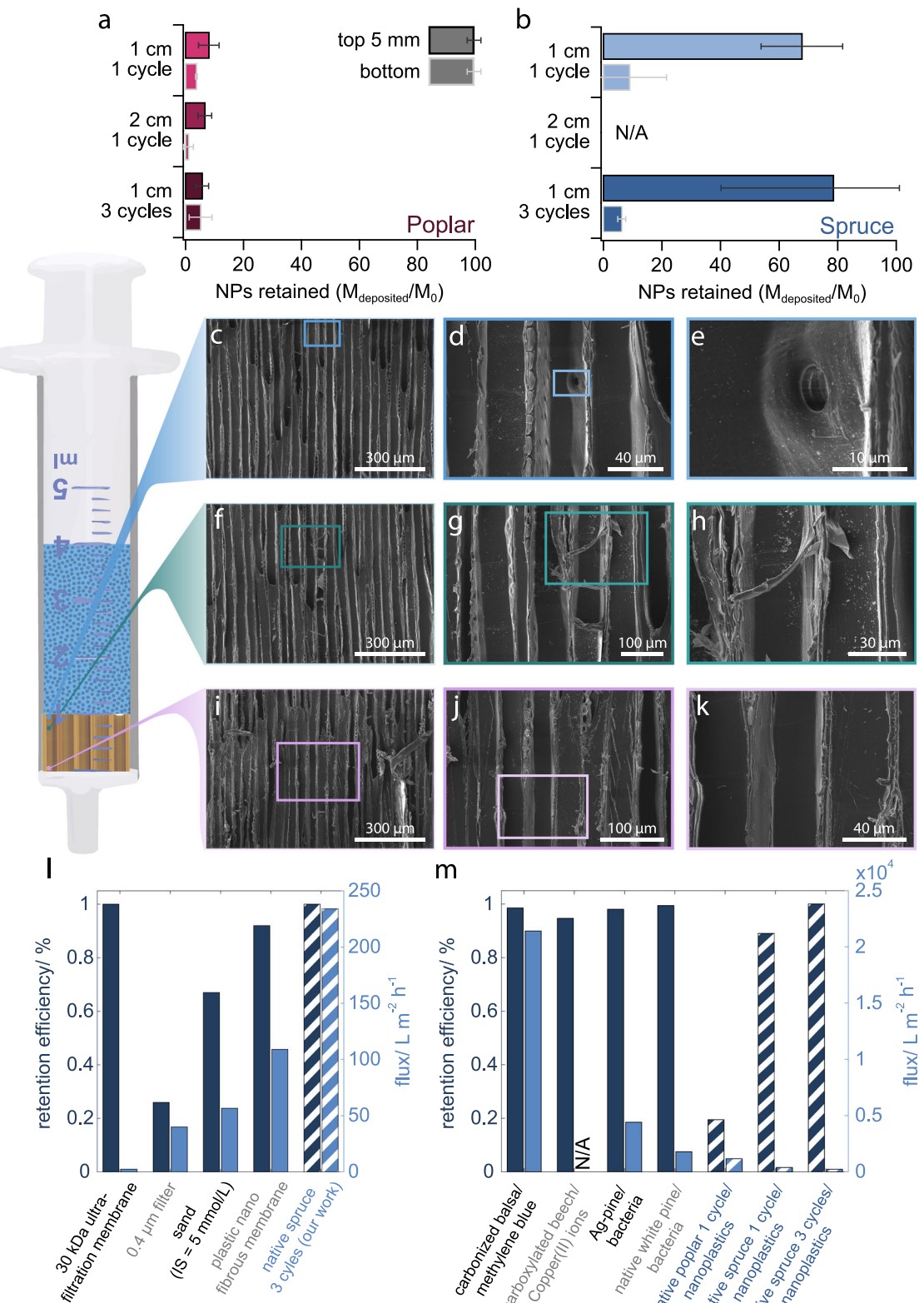

**Fig. 2 | Characterisation of Pd-NPs retention in the wood membranes, and comparison to retention efficiency of other filtration methods.** Ratio of the total Pd-NPs injected to those retained in top 5 mm and remaining bottom part of **a** poplar and **b** spruce filters, respectively, across the different thicknesses and filtration cycles ($n = 3$, error bar = standard deviation). Representative SEM micrographs taken at different positions of a spruce membranes: **c–e** at a depth of 0–1 mm from the top surface, where a pit is clearly shown, (**f–h**) at a depth of 2–3 mm, and (**i–l**) at a depth of 8–10 mm, corresponding to the bottom region of the filter.

Additional micrographs focusing on the area surrounded by the dotted line in (**d**) can be found in Figure S1. **l** Comparison of different NPs filtration methods (from left to right[15,32,40]:) with our work (Dashed bars) **m** Comparison of with different wood filtration applications (from left to right[26,41–43]:) with our work (Dashed bars), specifying the nature of the wood membrane followed by the contaminant removed. For each study in **l** and **m**, the left bar = retention efficiency, and the right bar = Flux. IS = ionic strength.

permeability is conducive to a higher retention efficiency of Pd-NP. Overall, although native spruce membranes have a lower flux than other wood membranes, their retention efficiency and flux are comparable with advanced filters while being environmentally-friendly, since they do not require chemical modification or energy-intensive fabrication.

Prior to deployment of this technology, further studies should investigate whether softwood membranes are also efficient at removing other colloidal contaminants[44,45]. For this, we suggest to study the effect of pit opening size on the retention efficiency of contaminants with different sizes and compositions[31,46]. Future research should also evaluate the scalability of wood membranes for real-world applications by quantifying their technical performance, including metrics such as transmembrane pressure, total filtration capacity before clogging, the potential for NP recovery, and membrane regeneration through backwashing. To complement this, life-cycle and techno-economic assessments will be essential to support their integration into sustainable water treatment technologies.

In conclusion, since it is anticipated that (nano)plastic contamination could be present in a number of drinking water sources, treatment may be necessary to reduce exposure. Such, water treatment processes should be both efficient and sustainable. Thanks to their anatomy (uniform tracheids of finite length connected laterally by pits), softwoods can efficiently remove NPs even without chemical modifications by retaining NPs around the pits and on the tracheid cell walls. Therefore, wood has the potential to be used as a membrane filter to provide safe drinking water at a reduced economic and environmental cost.

## Methods

### Preparation of Pd-NPs feed suspensions in artificial freshwater

MilliQ (Merck Millipore, $18.2\,M\Omega$) water was used throughout the experiments. Palladium (Pd) - labeled NPs (Pd-NPs) were synthesized and characterized according to a previously reported protocol[30]. The Pd-NPs featured a polyacrylonitrile core with embedded Pd encased in a polystyrene shell with a final dopant concentration of $0.295 \pm 0.009$ w/w % Pd. The initial hydrodynamic diameter and zeta-potential of the Pd-NPs were $210 \pm 19$ nm and $-52.83 \pm 2.23$ mV, respectively, as determined by dynamic light scattering and electrophoretic mobility measurements using the Smoluchowski model (Zetasizer Nano, Malvern Panalytical).

The Pd-NPs stock suspension was diluted by a factor $10^3$ with artificial soft freshwater to a final concentration of $199.47 \pm 5.99\,\mu g\,Pd\,L^{-1}$ and $67.637 \pm 0.15$ mg Pd-NPs $L^{-1}$. While such concentrations are orders of magnitude higher than expected in natural drinking water sources, they were selected ensure that concentrations of Pd-NPs in the effluent could be quantified. The artificial soft fresh water with an ionic strength of $11.4\,mmol\,L^{-1}$ was prepared according to US EPA guidelines with $48.0\,mg\,L^{-1}$ $NaHCO_3$ ($NaHCO_3$, 99.5–100%, Sigma-Aldrich), $30\,mg\,L^{-1}$ $Ca_2SO_4\,2H_2O$ ($CaSO_4{\cdot}2H_2O$, 98%, Sigma-Aldrich), $30\,mg\,L^{-1}$ $MgSO_4$ ($MgSO_4$, ≥99.5%, Sigma Aldrich), and $2\,mg\,L^{-1}$ KCl (KCl, ≥ 99.5%, Sigma Aldrich)[47]. In artificial freshwater, the Pd-NPs hydrodynamic diameter was $234 \pm 7$ nm, and the zeta-potential was $-40.75 \pm 0.49$.

### Pressure-driven wood membrane filtration

Cross sections of native Norway spruce (*Picea abies*) and Poplar (*Populus nigra*) wood were sawed to the required thicknesses of 10 mm and 20 mm. Round cross sections were prepared using a punching tool with a diameter of 14 mm. The wood membrane filter was placed at the bottom of a 12 mL syringe, with an effective membrane area of $154\,mm^2$. A thorough seal between the wood membrane and the inside wall of the syringe was ensured using a small amount of low viscosity cyanoacrylate glue. A custom holder accommodating four modified syringes was constructed to enable placement of collection vials beneath the outlets for recovery of the filtrate. The holder also allowed weights to be applied to each syringe plunger to push the dispersion through the membrane. The applied load was selected based on the effective filter area such that the resulting pressure corresponded to a constant pressure of 1 bar. Prior to the Pd-NPs filtration, the wood membrane was thoroughly rinsed with water. The Pd-

NPs feed suspension was re-dispersed for 10 s on a vortex mixer (Vortex Genie 2), followed by 15 min sonication. Each filtration run started by adding 3 mL Pd-NPs feed suspension to the syringe and applying approximately 1 bar pressure. In addition to single filtration cycles, we also analyzed the effect of three consecutive filtration cycles through the same wood membrane. Membranes were not dried between cycles, allowing conditions to remain representative of continuous filtration. The filtered solutions were collected and stored in a fridge before their concentration was measured by ICP-MS analysis (Supplementary Section 1). The wood membrane samples were retained and sliced at different depths to assess the content of Pd-NPs deposited, thereby also enabling a mass balance of Pd-NPs across the system.

### Scanning Electron Microscopy (SEM)

Wood membrane samples were pre-cut to reveal the tangential wood plane using a razor blade, followed by a subsequent surface polishing using a rotary microtome (Leica RM2255 with a Feather N35 microtome blade). The samples were then glued to a SEM stub and sputter-coated with 5 nm carbon using a Safematic CCU-010 coater. The coated top surface and the stub were connected using silver ink to allow for an electronic connection. Micrographs were obtained using a scanning electron microscope (Hitachi SU5000) using an acceleration voltage of 5 kV and under high vacuum. Elemental content distribution mapping was attempted by energy dispersive X-ray spectroscopy (EDS) spectrum imaging with an Oxford Ultim Max 100 EDS system attached to a Hitachi SU5000 SEM.

### Inductively Coupled Plasma—Mass Spectroscopy (ICP-MS)

To quantify NPs, samples were first concentrated, digested, and then analyzed by ICP-MS. Following the protocols used in previous studies[48,49], Pd-NPs in the filtrate were pre-concentrated by flocculation using poly(acrylamide co-acrylic acid) (CAS. 9003-06-9, Sigma-Aldrich) and aluminum nitrate nonahydrate (CAS 7784-27-2, Sigma-Aldrich). The protocol, which had a recovery rate of $95 \pm 5\%$, is further described in Supplementary Section 1. Wood membranes were cut into 4 pieces prior to digestion. Floccs and wood sections underwent microwave acid digestion in an ultraWAVE digestor (Milestone Srl, Italy) with either 2.3 mL or 5 mL distilled $HNO_3$ 65% (puriss. Sigma-Aldrich), respectively, for each of the sample types. The pressure and temperature increased from ambient to 220 °C and 160 bar over 30 min and were then maintained at 220 °C and 160 bar for 30 min. Digestates were then diluted, and $^{106}Pd$ was quantified by ICP-MS (Agilent 7900). Internal standards of Scandium, Yttrium, and Rhodium (10 μg/L) were added to the solutions and were continuously monitored throughout the measurements, where results were automatically corrected to account for instrumental drift.

## Data availability

The mass of Pd-NP eluted from wood membranes and the mass of Pd-NP deposited in wood membranes are available at https://doi.org/10.5281/zenodo.18890309. The following data repository: https://doi.org/10.5281/zenodo.18890309 contains the mass of Pd-NP eluted from wood membranes and the mass of Pd-NP deposited in wood membranes.

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

## Acknowledgements

W.Y. acknowledges funding from an ETH Zurich Career Seed Grant 1-009041. A.P. was funded by the ETH Postdoctoral Fellowship and the Rütli Foundation. D.M.M. was funded through the Swiss National Science Foundation (SNSF), grant number PCEFP2_186856. M.R. thanks Katharina Heider for the invaluable discussions on micro- and nanoplastic contamination, which served as inspiration for this work. We acknowledge Dan Vivas Glaser for his help with sample preparation and Thomas Schnider for wood cutting. We thank Prof. Dr. Ingo Burgert (ETH Zürich) for his continuous support.

## Author contributions

M.R. and A.P. contributed equally to this work. W.Y., M.R., A.P., and D.M.M. contributed to the conception of the study, the experimental design, and the writing of the manuscript. W.Y., M.R., and A.P. were responsible for the planning, organization, and execution of experiments.

## Competing interests

The authors declare no competing interests.
