## [Transparent Peer Review file · Communications Earth & Environment]

Water filtration using softwood membranes provides a nature-based solution for nanoplastic removal

Corresponding Author: Dr Alice Pradel

Version 0:

Decision Letter:

Dear Dr Pradel,

Your manuscript titled "Nanoplastic filtration using native wood membranes: A nature-based solution to water purification" has now been seen by 3 reviewers, and we include their comments at the end of this message. They find your work of interest, but some important points are raised. We are interested in the possibility of publishing your study in Communications Earth & Environment, but would like to consider your responses to these concerns and assess a revised manuscript before we make a final decision on publication.

We therefore invite you to revise and resubmit your manuscript, along with a point-by-point response that takes into account the points raised. Please highlight all changes in the manuscript text file.

Please submit your point-by-point responses as a separate file, distinct from your cover letter where you can add responses to the Editors' comments that you do not want to be made available to the reviewers. Word files are preferred. We recommend that any figures, tables or graphs that are included in the response to reviewers are also included in the main article or Supplementary Information.

Please use the following link to submit your revised manuscript, point-by-point response to the referees' comments (which should be in a separate document to any cover letter), a tracked-changes version of the manuscript (as a PDF file) and the completed checklist:

Link Redacted

We hope to receive your revised paper within six weeks; please let us know if you aren't able to submit it within this time so that we can discuss how best to proceed. If we don't hear from you, and the revision process takes significantly longer, we may close your file. In this event, we will still be happy to reconsider your paper at a later date, as long as nothing similar has been accepted for publication at Communications Earth & Environment or published elsewhere in the meantime.

Please do not hesitate to contact us if you have any questions or would like to discuss these revisions further. We look forward to seeing the revised manuscript and thank you for the opportunity to review your work.

Best regards,

Somaparna Ghosh, PhD
Associate Editor,
Communications Earth & Environment
Consulting Editor,
Communications Sustainability

EDITORIAL POLICIES AND FORMATTING

- Behavioural and social science
- Ecological, evolutionary & environmental sciences
- Life sciences

Furthermore, please align your manuscript with our format requirements, which are summarized on the following checklist: <https://www.nature.com/documents/commsj-phys-style-formatting-checklist-article.pdf> Communications Earth & Environment formatting checklist

and also in our style and formatting guide <https://www.nature.com/documents/commsj-phys-style-formatting-guide-accept.pdf> Communications Earth & Environment formatting guide .

***** DATA:** Communications Earth & Environment endorses the principles of the Enabling FAIR data project (<http://www.copdess.org/enabling-fair-data-project/>). We ask authors to make the data that support their conclusions available in permanent, publically accessible data repositories. (Please contact the editor if you are unable to make your data available).

All Communications Earth & Environment manuscripts must include a section titled "Data Availability" at the end of the Methods section or main text (if no Methods). More information on this policy, is available at <http://www.nature.com/authors/policies/data/data-availability-statements-data-citations.pdf>

If a community resource is unavailable, data can be submitted to generalist repositories such as <https://figshare.com/> or <http://datadryad.org/> Dryad Digital Repository. Please provide a unique identifier for the data (for example a DOI or a permanent URL) in the data availability statement, if possible. If the repository does not provide identifiers, we encourage authors to supply the search terms that will return the data. For data that have been obtained from publically available sources, please provide a URL and the specific data product name in the data availability statement. Data with a DOI should be further cited in the methods reference section.

REVIEWER COMMENTS:

Reviewer #1 (Remarks to the Author):

This work investigates the use of softwood and hardwood for the filtration of nanoplastic of 1 nm to 1 micron size. The result is encouraging as nanoplastic can indeed be retained using these wood with structure that can retain the nanoplastic. This is a well written short communication. Nevertheless, the introduction of membrane technology can be improved especially the introduction of current established membranes such as ultrafiltration, nanofiltration and reverse osmosis membranes. Tight ultrafiltration (typically pore size of few nanometers could also be used for such application). Some suggestions are provided to help improve the manuscript.

1) Line 51, Introduction: It is true that GO and MOF are still distance, but current pressure-driven membrane technology (UF/NF/RO) work well for NPs of 1 nm to 1 micron size. Suggest introducing pressure-driven membrane processes in the removal of microplastics and nanoplastics. It is also worth to mention that current membrane production also releases NPs to

the environment to justify the look for other alternative tech/ materials. And thus, the use of natural materials such as wood for filtration could provide a potential alternative solution.

2) It was not clear why the use of these wood membranes with tracheids of diameter 30 to 50 micron could remove NP (1 nm to 1 micron). In the manuscript, it is mentioned that the pits with unknown channel size is the main reason to the particle retention. The pore size distribution of the hardwood and softwood should be characterized and compared with the structure of the wood. Suggest performing liquid porosimetry or nitrogen sorption experiment to obtain pore size distribution and potentially the pit size.

3) Line 123: The membrane performance should be reported in terms of pure water flux (L/(m²/h)) for comparison purpose instead of time taken to pass through the same amount of feed. It would also be good to test the membrane performance over extended period of time to determine the stability of the structure.

4) Figure 2i,m: Both flux and rejections are important indicators for membranes and should be considered when comparing with the other membranes. The membranes for comparison should be based on common, established membranes such as polysulfone or polyamide membranes rather than 2D nanosheets which are not used in practical applications.

Reviewer #2 (Remarks to the Author):

The article describes a simple yet innovative method for the sequestration of nanoplastics dispersed in artificial freshwater solutions prepared in laboratory. Membranes of different thicknesses from representative species of both softwood (spruce) and hardwood (poplar) categories are compared. From this preliminary investigation, the membrane from spruce, thanks to its cellular structure, promotes better retention of Pd-NP compared to hardwood. The experimental investigations performed, ICP-MS and SEM, support the conclusions indicated by the authors.

The work appears to be well written, the methods are reported accurately, and in its completeness the work is very interesting.

However, the authors should clarify some elements in order to make the work more complete:

1. Clarify the method used for monitoring and maintaining the pressure of approximately 1 bar reported in the "Pressure-Driven Wood Membrane Filtration" section
2. The authors state that it was not possible to perform investigations by EDS due to the low concentration of Pd in the Pd-NPs. It would be necessary to consider alternative analytical techniques to support the confirmation of palladium presence within the membranes.

Reviewer #3 (Remarks to the Author):

Review Comments on

Nanoplastic filtration using native wood membranes: A nature-based solution to water purification

For: Communications Earth & Environment

The paper reports on proof-of-concept work on using wood membranes (literally cross-sectional slices of wood) to remove nano-plastics from water.

The grammar and sentence structure are excellent. No problems with the writing. The method and materials used were clearly described and the results presented in a clear and logical way.

The matrix of experiments includes two wood species, and two membrane thicknesses. A synthetic mono-disperse particle size (210nm) was used. The dispersion of particles within the wood was characterized by SEM and EDAX. The particle concentration in the retentate and permeate was determined by ICP-MS (which detected the palladium tracer in the particles).

The results were impressive for the spruce wood, with 90% of the nanoparticles being retained (rejected) in a single pass and 99% in three passes. The poplar membrane only achieved 20% retention.

No major issues

Minor issues and comments:

In my experience membrane performance is measured by the retention rate (aka rejection rate) of some specific molecule or particle of interest. In several places the authors switch to the term "transport rate," which is fraction that passes through. This is not really useful to the reader and just creates confusion. They should stick to retention rate as the performance parameter.

Trans-membrane pressure (TMP) and permeate flux are key pieces of information in all membrane experiments. For future work I recommend measuring and reporting the pressure inside your membrane test cell. Also, a plot of flux versus TMP over a reasonable range of pressures for your system would be helpful to get an idea of the membrane area required for a typical application.

They should mention if they envision these wood disk membranes would go through a cleaning cycle once they get fouled, or simply discarded. A slice of wood can't be very expensive. Cleaning may not be worth the effort.

In the future work section, I would add testing of other wood species, and covering a range of particle sizes. Also, would heating the wood alter the pore size?

Lastly, for future experiments, they might consider using a more flexible adhesive to seal around the membrane. Cyanoacrylate adhesives are quite brittle. Perhaps look at the polyurethane options.

Recommendation is to publish with minor corrections.

I do hope they continue this work. These wood membranes could be a very cost-effective option for water treatment.

Scott Siquefield PhD
GA Tech

** Visit Nature Portfolio's author and referees' website at www.nature.com/authors for information about policies, services and author benefits**

Communications Earth & Environment is committed to improving transparency in authorship. As part of our efforts in this direction, we are now requesting that all authors identified as 'corresponding author' create and link their Open Researcher and Contributor Identifier (ORCID) with their account on the Manuscript Tracking System prior to acceptance. ORCID helps the scientific community achieve unambiguous attribution of all scholarly contributions. You can create and link your ORCID from the home page of the Manuscript Tracking System by clicking on 'Modify my Springer Nature account' and following the instructions in the link below. Please also inform all co-authors that they can add their ORCIDs to their accounts and that they must do so prior to acceptance.

Version 1:

Decision Letter:

Dear Dr Pradel,

Your manuscript titled "Nanoplastic filtration using native wood membranes: A nature-based solution to water purification" has now been seen by our reviewers, whose comments appear below. In light of their advice we are delighted to say that we are happy, in principle, to publish a suitably revised version in Communications Earth & Environment.

We therefore invite you to revise your paper one last time to address the remaining concerns of our reviewers. At the same time we ask that you edit your manuscript to comply with our format requirements and to maximise the accessibility and therefore the impact of your work.

EDITORIAL REQUESTS:

*****Please take care to match our formatting and policy requirements. We will check revised manuscript and return manuscripts that do not comply. Such requests will lead to delays. *****

SUBMISSION INFORMATION:

OPEN ACCESS:

Communications Earth & Environment is a fully open access journal. Articles are made freely accessible on publication. For further information about article processing charges, open access funding, and advice and support from Nature Portfolio, please visit <https://www.nature.com/commsenv/open-access>

Link Redacted

Best regards,

Somaparna Ghosh, PhD
Associate Editor,
Communications Earth & Environment
Consulting Editor,
Communications Sustainability

REVIEWERS' COMMENTS:

Reviewer #1 (Remarks to the Author):

I am happy with the revision.
Recommendation: Accept.

Reviewer #2 (Remarks to the Author):

After examination of the reviewers' comments, the authors' responses, and the corresponding revisions to the manuscript, I find the revisions to be satisfactory and therefore recommend publication of the article.

Reviewer #3 (Remarks to the Author):

The paper reports on proof-of-concept work on using wood membranes (literally cross-sectional slices of wood) to remove nano-plastics from water.

The authors have addressed all of the reviewer's comments adequately and I recommend publication of this paper. We covered the major claims in the first review. This is the final review. The paper presents a novel idea in the field and will influence thinking.

** Visit Nature Portfolio's author and referees' website at <http://www.nature.com/authors> for information about policies, services and author benefits**

REVIEWER COMMENTS:

Reviewer #1 (Remarks to the Author):

This work investigates the use of softwood and hardwood for the filtration of nanoplastic of 1 nm to 1 micron size. The result is encouraging as nanoplastic can indeed be retained using these wood with structure that can retain the nanoplastic. This is a well written short communication. Nevertheless, the introduction of membrane technology can be improved especially the introduction of current established membranes such as ultrafiltration, nanofiltration and reverse osmosis membranes. Tight ultrafiltration (typically pore size of few nanometers could also be used for such application). Some suggestions are provided to help improve the manuscript.

1) Line 51, Introduction: It is true that GO and MOF are still distance, but current pressure-driven membrane technology (UF/ NF/ RO) work well for NPs of 1 nm to 1 micron size. Suggest introducing pressure-driven membrane processes in the removal of microplastics and nanoplastics. It is also worth to mention that current membrane production also releases NPs to the environment to justify the look for other alternative tech/ materials. And thus, the use of natural materials such as wood for filtration could provide a potential alternative solution.

Thank you for this suggestion. We have made such improvements:

Lines 63-68 : Current pressure-driven membrane technology, such as ultra-filtration, nano-filtration and reverse osmosis can retain colloidal contaminants, such as NPs, by size exclusion¹⁻³. However, as they are composed of synthetic polymers, their production and subsequent use may release NPs, and so the use of alternative, bio-sourced, biodegradable materials could alleviate some of these concerns. For example, a bio-based nanofibre hydrogel has shown promise to remove NPs larger than 30 nm⁴.

2) It was not clear why the use of these wood membranes with tracheids of diameter 30 to 50 micron could remove NP (1 nm to 1 micron). In the manuscript, it is mentioned that the pits with unknown channel size is the main reason to the particle retention. The pore size distribution of the hardwood and softwood should be characterized and compared with the structure of the wood. Suggest performing liquid porosimetry or nitrogen sorption experiment to obtain pore size distribution and potentially the pit size.

In the revised version of the manuscript, we have more clearly explained the process by which Pd-NPs are retained in constrictions with diameters larger than the Pd-NPs. We provided the hypothesis that it is the presence of constrictions, created by the pits, that enables deposition and subsequent retention of the Pd-NPs (Fig S4). Indeed, according to colloid filtration theory, retention of nanoparticles in porous media can occur by interception and diffusion. Interception of colloids at constriction zones (i.e., pores) that are larger than the colloid size is commonly observed by processes of bridging or progressive build-up of deposits (i.e., ripening).

While we have not performed liquid porosimetry, we have found studies that performed such measurements on Norway spruce and have now cited them accordingly. The size distributions described therein are in agreement with our SEM observations.

Lines 161-168 : The openings of these pits, which are 3 to 7 μm in diameter, are smaller than the tracheid diameter (20 to 50 μm , depending on the wood's age). As a result, they act as constriction zones, increasing the likelihood of interception and deposition on the wood surface^{5,6} (Fig. S4). In this case, Pd-NPs are not retained by size exclusion (i.e., pores smaller than the colloid size), but rather by interception around the pit. This occurs due to the change in streamline direction at the constriction formed by the pit. Such interception at constrictions (i.e.: pore throats) with diameters larger than the colloid, is suspected to occur in packed-bed filters^{7,8}, though it cannot be observed directly. However, this process has been observed in microfluidic filters⁹⁻¹¹.

3) Line 123: The membrane performance should be reported in terms of pure water flux ($\text{L}/(\text{m}^2/\text{h})$) for comparison purpose instead of time taken to pass through the same about of feed. It would also be good to test the membrane performance over extended period of time to determine the stability of the structure.

We have moved the membrane performance in terms of pure water flux from the materials and methods section to the main part of the manuscript in the revised version of the text.

Lines 170-178: These findings are further supported by observations that with the same applied pressure, spruce and poplar membranes exhibited different water fluxes. Based on the filtration volume (3 mL), membrane area (154 mm^2), and approximate filtration durations (57 seconds to 5 min), the water flux values were $608 \text{ L m}^{-2} \text{ h}^{-1} \text{ bar}^{-1}$ for poplar (after 1 or 3 filtrations) and 195 and 117 $\text{L m}^{-2} \text{ h}^{-1} \text{ bar}^{-1}$ for spruce during the first and third filtrations, respectively. Corresponding flowrates were 1.2 m h^{-1} for poplar and 0.4 m h^{-1} and 0.2 m h^{-1} for spruce. These values, obtained for spruce tracheids of 20 to 50 μm diameter and poplar vessels of approximately 150 μm diameter. For reference, the flowrate for poplar is slightly above that of typical slow sand filtration but considerably lower than rapid sand filtration (7–20 m h^{-1}) in a conventional drinking water treatment plant while the flowrate for spruce is in the range of slow sand filtration ($<0.3 \text{ m h}^{-1}$).

This has also been discussed further in Figure 2m and lines 319 to 340 of the revised manuscript.

4) Figure 2i,m: Both flux and rejections are important indicators for membranes and should be considered when comparing with the other membranes. The membranes for comparison should be based on common, established membranes such as polysulfone or polyamide membranes rather than 2D nanosheets which are not used in practical applications.

Lines 322 - 341 : *On one hand, native spruce membranes outperformed many ultrafiltration, microfiltration and packed bed filtration systems^{8,12}. Spruce membranes achieved equal retention efficiency as ultrafiltration membranes, but at a significantly higher flow rate¹². They could achieve higher retention efficiency than filter microfiltration membranes (0,4 μm)¹² and sand filters⁸, and they could also achieve higher flow rates. Finally, nanofibrous membranes composed of synthetic polymers also achieve retention efficiencies up to approximately 0.9, but these had a lower flux compared to spruce membranes, and require energy- and material-intensive preparations³ (**Figure 2l**). On the other hand, the filtration of NPs measured in our work falls within the ranges of retention efficiencies and fluxes observed for other woods (native or functionalized) and other contaminants (dissolved or bacterial; **Figure 2m**). To achieve over 90 % retention efficiencies of organic molecules (methylene blue) and metallic ions (copper), wood membranes need to be functionalized^{13,14}. Filtration of methylene blue by carbonized wood achieved the highest combination of retention efficiency and flux¹³. Bacteria were the only particulate matter whose filtration was studied^{15,16}. Results showed that both functionalized and native (i.e.: non-functionalized) wood achieved well above 90% retention of bacteria^{15,16}, but the fluxes were significantly lower than for methylene blue in balsa wood¹³. At the same filtration pressure, the flux in both our spruce and poplar membranes was lower than the flux in bacterial filtration studies. Furthermore, spruce membranes, which had a higher retention efficiency of Pd- NP compared to poplar membranes, also had a lower flux compared to poplar membranes (**Figure 2m**). This suggests that a low wood membrane permeability is conducive to a higher retention efficiency of Pd-NP. Overall, although native spruce membranes have a lower flux than other wood membranes, their retention efficiency and flux is comparable with advanced filters while being environmentally-friendly, since they do not require any modifications.*

Reviewer #2 (Remarks to the Author):

The article describes a simple yet innovative method for the sequestration of nanoplastics dispersed in artificial freshwater solutions prepared in laboratory. Membranes of different thicknesses from representative species of both softwood (spruce) and hardwood (poplar) categories are compared. From this preliminary investigation, the membrane from spruce, thanks to its cellular structure, promotes better retention of Pd-NP compared to hardwood. The experimental investigations performed, ICP-MS and SEM, support the conclusions indicated by the authors.

The work appears to be well written, the methods are reported accurately, and in its completeness the work is very interesting.

However, the authors should clarify some elements in order to make the work more complete:

1. Clarify the method used for monitoring and maintaining the pressure of approximately 1 bar reported in the "Pressure-Driven Wood Membrane Filtration" section

We unfortunately do not have a photograph of the system but have described it in more details in the revised version of the manuscript.

Lines 447-451: A custom holder accommodating four modified syringes was constructed to enable placement of collection vials beneath the outlets for recovery of the filtrate. The holder also allowed weights to be applied to each syringe plunger, to push the dispersion through the membrane. The applied load was selected based on the effective filter area such that the resulting pressure corresponded to a constant pressure of 1 bar.

2. The authors state that it was not possible to perform investigations by EDS due to the low concentration of Pd in the Pd-NPs. It would be necessary to consider alternative analytical techniques to support the confirmation of palladium presence within the membranes.

We chose SEM-EDS since it allows to combine imaging and chemical mapping of a sample. Other techniques, such as laser ablation ICP-MS and imagine mass cytometry, only enable chemical mapping. In a previous study, the Pd content of Pd-NPs was successfully observed by TEM-EDS using 40 keV energy and a spot-size of 6, combined with software-enhancement of chemical signal¹⁷. These analytical conditions were not possible in our case, since i) SEM was necessary to observe the wood structure instead of TEM and ii) the EDS energy had to be lowered to avoid damaging the wood. Hence, we have streamlined our explanation of why Pd-NPs were not visible by SEM-EDS.

Lines 288-291 : “We attempted to confirm the presence of deposited Pd-NPs in the wood membranes by measuring Pd by EDS, as done previously¹⁷, but this was not achievable since the high acceleration voltage necessary for obtaining a Pd signal resulted in beam damage of the wood.”

Despite not being able to confirm Pd-content, the unique morphology of the model Pd-NPs is clearly recognizable in the SEM images (Fig. 2c-k and Fig. S1, S3, S4) and is very well correlated to the quantifications of Pd-NP deposited in different sections of the membrane measured by ICP-MS. Therefore, in the updated version of the manuscript, we have now clarified how these complementary techniques allowed us to confirm the presence of Pd-NPs in our test system.

Lines 273-275 : “These deposition profiles obtained by measuring Pd-NPs in cross sections of the wood membrane are in agreement with scanning electron microscope (SEM) micrographs taken at different depths of a spruce membrane”

Reviewer #3 (Remarks to the Author):

Review Comments on Nanoplastic filtration using native wood membranes: A nature-based solution to water purification For: Communications Earth & Environment

The paper reports on proof-of-concept work on using wood membranes (literally cross-sectional slices of wood) to remove nano-plastics from water.

The grammar and sentence structure are excellent. No problems with the writing. The method and materials used were clearly described and the results presented in a clear and logical way. The matrix of experiments includes two wood species, and two membrane thicknesses. A synthetic mono-disperse particle size (210nm) was used. The dispersion of particles within the wood was characterized by SEM and EDAX. The particle concentration in the retentate and permeate was determined by ICP-MS (which detected the palladium tracer in the particles).

The results were impressive for the spruce wood, with 90% of the nanoparticles being retained (rejected) in a single pass and 99% in three passes. The poplar membrane only achieved 20% retention.

No major issues

Minor issues and comments:

In my experience membrane performance is measured by the retention rate (aka rejection rate) of some specific molecule or particle or interest. In several places the authors switch to the term “transport rate,” which is fraction that passes through. This is not really useful to the reader and just creates confusion. They should stick to retention rate as the performance parameter.

Thank you for this suggestion. We now only use the term retention efficiency, that we have defined at the beginning of the results and discussion section:

Lines 137-139: The retention efficiency of Pd-NPs, obtained by dividing the change in Pd-NP concentration in the effluent ($C_0 - C$) by the initial concentration (C_0), provided a measure of the capacity of Pd-NP to be retained by wood membranes

Trans-membrane pressure (TMP) and permeate flux are key pieces of information in all membrane experiments. For future work I recommend measuring and reporting the pressure inside your membrane test cell. Also, a plot of flux versus TMP over a reasonable range of pressures for your system would be helpful to get an idea of the membrane area required for a typical application.

We have now calculated the permeate flux and discussed it in the manuscript lines 170-178 and lines 322 to 341 (see response to reviewer 1, pages 2 and 3)

We have not measured trans-membrane pressure, but we have mentioned that a constant pressure of 1 bar is applied in materials and methods section.

Lines 450-451: The applied load was selected based on the effective filter area such that the resulting pressure corresponded to a constant pressure of 1 bar.

Furthermore, we have added that trans-membrane pressure is a parameter to be studied in order to scale wood membranes for larger-scale applications.

Lines 346-349: Future research should also evaluate the scalability of wood membranes for real-world applications by quantifying their technical performance, including metrics such as transmembrane-pressure, total filtration capacity before clogging, the potential for NP recovery, and membrane regeneration through backwashing.

They should mention if they envision these wood disk membranes would go through a cleaning cycle once they get fouled, or simply discarded. A slice of wood can't be very expensive. Cleaning may not be worth the effort.

We have not tested the effectiveness of cleaning the wood membranes by backwashing. This is worth testing to assess whether wood membranes can be re-used. We have added this consideration to the manuscript lines 340-343 (cf: previous comment). Furthermore, the cost analysis is mentioned in the following sentence :

Lines 349-350 : To complement this, life-cycle and techno-economic assessments will be essential to support their integration into sustainable water treatment technologies.

In the future work section, I would add testing of other wood species, and covering a range of particle sizes. Also, would heating the wood alter the pore size?

Heating would not affect the pore size. Some methods exist to dissolve the cell-wall and re-precipitate it inside the lumen, but these are too destructive to the wood's mechanical integrity to be applicable to filtering. Testing different tree species is worthwhile to study the effect of pit opening size in future work.

Lines 344-345 : For this, we suggest to study the effect of pit opening size on the retention efficiency of contaminants with different sizes and compositions^{5,6}.

Lastly, for future experiments, they might consider using a more flexible adhesive to seal around the membrane. Cyanoacrylate adhesives are quite brittle. Perhaps look at the polyurethane options.

Thank you for your suggestion. We will consider this in future work. We envision that the type of seal used would change when the membrane is used at larger scales.

Recommendation is to publish with minor corrections.

I do hope they continue this work. These wood membranes could be a very cost-effective option for water treatment.

Thank you for your encouraging feedback!

Scott Siquefield PhD
GA Tech

References :

1. Li, M., Huang, G., Wang, S., Huang, J. & Xu, Z. Development of hydroxyapatite-enhanced membrane for nanoplastics removal: Multiple scenarios and mechanism exploration. *J. Hazard. Mater.* **495**, 138781 (2025).
2. Bodzek, M. & Pohl, A. Possibilities of removing microplastics from the aquatic environment using membrane processes. *Desalination Water Treat.* **288**, 104–120 (2023).
3. Wan, H. *et al.* Removal of polystyrene nanoplastic beads using gravity-driven membrane filtration: Mechanisms and effects of water matrices. *Chem. Eng. J.* **450**, 138484 (2022).
4. Jiang, M. *et al.* A bio-based nanofibre hydrogel filter for sustainable water purification. *Nat. Sustain.* **7**, 168–178 (2024).
5. Patera, A., Bonnin, A. & Mokso, R. Micro- and Nano-Scales Three-Dimensional Characterisation of Softwood. *J. Imaging* **7**, 263 (2021).
6. Plötze, M. & Niemz, P. Porosity and pore size distribution of different wood types as determined by mercury intrusion porosimetry. *Eur. J. Wood Wood Prod.* **69**, 649–657 (2011).
7. Bradford, S. A., Torkzaban, S. & Walker, S. L. Coupling of physical and chemical mechanisms of colloid straining in saturated porous media. *Water Res.* **41**, 3012–3024 (2007).
8. Pradel, A. *et al.* Deposition of environmentally relevant nanoplastic models in sand during transport experiments. *Chemosphere* **255**, 126912 (2020).
9. Pradel, A., Delouche, N., Gigault, J. & Tabuteau, H. Role of Ripening in the Deposition of Fragments: The Case of Micro- and Nanoplastics. *Environ. Sci. Technol.* **58**, 8878–8888 (2024).
10. Vani, N., Escudier, S., Jeong, D.-H. & Sauret, A. Role of the constriction angle on the clogging by bridging of suspensions of particles. *Phys. Rev. Res.* **6**, L032060 (2024).
11. Okaybi, W., Roman, S. & Soulaire, C. Progressive colloidal clogging mechanism by dendritic build-up in porous media. *Soft Matter* **21**, 5687–5698 (2025).
12. Molina, S., Ocaña-Biedma, H., Rodríguez-Sáez, L. & Landaburu-Aguirre, J. Experimental Evaluation of the Process Performance of MF and UF Membranes for the Removal of Nanoplastics. *Membranes* **13**, 683 (2023).
13. Jiao, M. *et al.* Highly Efficient Water Treatment via a Wood-Based and Reusable Filter. *ACS Mater. Lett.* **2**, 430–437 (2020).
14. Vitas, S., Keplinger, T., Reichholf, N., Figi, R. & Cabane, E. Functional lignocellulosic material for the remediation of copper(II) ions from water: Towards the design of a wood filter. *J. Hazard. Mater.* **355**, 119–127 (2018).
15. Boutilier, M. S. H., Lee, J., Chambers, V., Venkatesh, V. & Karnik, R. Water Filtration Using Plant Xylem. *PLoS ONE* **9**, e89934 (2014).
16. Che, W. *et al.* Wood-Based Mesoporous Filter Decorated with Silver Nanoparticles for Water Purification. *ACS Sustain. Chem. Eng.* **7**, 5134–5141 (2019).
17. Mitrano, D. M. *et al.* Synthesis of metal-doped nanoplastics and their utility to investigate fate and behaviour in complex environmental systems. *Nat. Nanotechnol.* **14**, 362–368 (2019).

Reviewer #1 (Remarks to the Author):

I am happy with the revision.
Recommendation: Accept.

Thank you for your final revision.

Reviewer #2 (Remarks to the Author):

After examination of the reviewers' comments, the authors' responses, and the corresponding revisions to the manuscript, I find the revisions to be satisfactory and therefore recommend publication of the article.

Thank you for your final revision.

Reviewer #3 (Remarks to the Author):

The paper reports on proof-of-concept work on using wood membranes (literally cross-sectional slices of wood) to remove nano-plastics from water.

The authors have addressed all of the reviewer's comments adequately and I recommend publication of this paper. We covered the major claims in the first review. This is the final review. The paper presents a novel idea in the field and will influence thinking.

Thank you for your final revision and kind words of support.